# Production and Characterization of Rhamnolipids Produced by *Pseudomonas aeruginosa* DBM 3774: Response Surface Methodology Approach

**DOI:** 10.3390/microorganisms10071272

**Published:** 2022-06-22

**Authors:** Olga Maťátková, Jana Michailidu, Richard Ježdík, Irena Jarošová Kolouchová, Tomáš Řezanka, Vladimír Jirků, Jan Masák

**Affiliations:** 1Department of Biotechnology, University of Chemistry and Technology, Technická 5, 166 28 Prague, Czech Republic; jana.michailidu@vscht.cz (J.M.); richardjezdik@gmail.com (R.J.); irena.kolouchova@vscht.cz (I.J.K.); vladimir.jirku@vscht.cz (V.J.); jan.masak@vscht.cz (J.M.); 2Institute of Microbiology, Czech Academy of Sciences, Vídeňská 1083, 142 20 Prague, Czech Republic; rezanka@biomed.cas.cz

**Keywords:** biosurfactants, fractional factorial design, *Pseudomonas aeruginosa*

## Abstract

Rhamnolipids are extensively studied biosurfactants due to their potential in many industrial applications, eco-friendly production and properties. However, their availability for broader application is severely limited by their production costs, therefore the optimization of efficacy of their cultivation gains significance as well as the information regarding the physio-chemical properties of rhamnolipids resulting from various cultivation strategies. In this work, the bioprocess design focused on optimization of the rhamnolipid yield of *Pseudomonas aeruginosa* DBM 3774 utilizing the response surface methodology (RSM). Six carbon sources were investigated for their effect on the rhamnolipid production. The RSM prediction improved the total rhamnolipid yield from 2.2 to 13.5 g/L and the rhamnolipid productivity from 11.6 to 45.3 mg/L/h. A significant effect of the carbon source type, concentration and the C/N ratio on the composition of the rhamnolipid congeners has been demonstrated for cultivation of *P. aeruginosa* DBM 3774 in batch cultivation. Especially, changes in presence of saturated fatty acid in the rhamnolipid congeners, ranging from 18.8% of unsaturated fatty acids (carbon source glycerol; 40 g/L) to 0% (sodium citrate 20 g/L) were observed. This demonstrates possibilities of model based systems as basis in cultivation of industrially important compounds like biosurfactants rhamnolipids and the importance of detailed study of interconnection between cultivation conditions and rhamnolipid mixture composition and properties.

## 1. Introduction

Biosurfactants are surface active compounds with varied structure, produced by a wide spectrum of microorganisms. These substances consist of a hydrophilic moiety (carboxylic acids, peptides, mono-, di-, or polysaccharides) and a hydrophobic moiety, usually comprising a saturated or unsaturated hydrocarbon or a fatty acid [1]. These compounds find their potential applications in bioremediation technologies, pharmaceutical industry, cosmetics, fine chemical and food manufacture [2].

*Pseudomonas aeruginosa* is one of the most studied bacterial producers of biosurfactants. This ubiquitous bacterium can be found in various environments and can also be advantageously used in biotechnological applications, such as microbial fuel cells or desalination cells [3,4]; however, the main focus is on the production of biosurfactants, rhamnolipids. These compounds are usually produced as a mixture of congeners, comprised of one or two molecules of rhamnose and one or two molecules of β-hydroxy fatty acid. These can also be accompanied by 3-hydroxyalkanoyloxy-alkanoates (HAAs) which are synthesized as part of the rhamnolipids biosynthetic pathway [5]. Briefly, the biosynthesis of rhamnolipids involves RhlA, which synthesizes 3-(3-hydroxyalkanoyloxy)alkanoic acids by esterification of two 3-hydroxyacyl molecules bound to acyl carrier protein from the fatty acid de novo synthesis. Subsequently, RhlB catalyzes reaction between an HAA molecule with dTDP-L-rhamnose descending from glucose-6-phosphate to form monorhamnolipids. In the third step, RhlC catalyzes the formation of dirhamnolipids [6]. The research reported by Wittgens*,* et al. [7] suggested possible mechanism how rhamno-monolipids are biosynthesized. They found that RhlB requires for its activity only HAA and thus cannot use single 3-hydroxyfatty acids for the synthesis of rhamno-monolipids. Therefore, these groups of rhamnolipids are probably products of subsequent RhaFAFA and RhaRhaFAFA hydrolysis, which is catalyzed by an enzyme from esterase or lipase family. Rhamnolipids are known for their excellent surface activities [8]. They can reduce the surface tension of water from 72 mN/m to 30 mN/m [9] at low critical micelle concentrations. Although rhamnolipids exhibit many activities that predestinate them for use in various industrial applications, the bottleneck of their commercial application is a low production yield and high production cost [1]. For this reason, many studies have been focused on finding a new efficient rhamnolipid producer and/or cost effective bioprocess settings [10,11]. It is known that the rhamnolipid production is influenced by many factors, including the type of carbon and nitrogen sources, culture conditions and many others. These factors can also influence the rhamnolipid congener composition and therefore the physico-chemical properties of the final product [12]. For rhamnolipid synthesis by *Pseudomonas aeruginosa*, one of the most important factors is the carbon source type and C/N ratio, as the production usually occurs at nitrogen limiting conditions [13]. The optimization of rhamnolipid production can be a time consuming process when individual factors are assessed separately [14]. The optimizable factors might include external conditions (agitation, aeration, pH, temperature, inoculums size) [15] but mostly the composition of the culture medium (the type and the concentration of the macro and micronutrients) as the main cultivation-related cost of production [14,16,17]. We have used a statistical optimization strategy based on response surface methodology (RSM) to determine the possibility of its use for optimization of the C/N content and ratio media composition for the enhancement of the rhamnolipid production by *Pseudomonas aeruginosa* DBM 3774 in batch cultivation. Our first goal was the evaluation of suitability of the response surface methodology for the optimization of the rhamnolipid yield in *Pseudomonas aeruginosa* DBM 3774. The novelty of the manuscript lies in the subsequent determination and comparison of the physico-chemical properties of rhamnolipids obtained under different cultivation conditions, especially with regard to evaluating the rhamnolipid congener composition and physio-chemical properties important for the biosurfactant applications, such as critical micelle concentration or solubilization of hydrophobic compounds.

## 2. Material and Methods

### 2.1. Microorganism

Rhamnolipid-producing bacterium *Pseudomonas aeruginosa* DBM 3774 was obtained from the Collection of Microorganisms of the Department of Biochemistry and Microbiology, University of Chemistry and Technology, Prague.

### 2.2. Cultivation in Erlenmeyer Flasks

The cultivation was carried out in 500 mL Erlenmeyer flasks (30 °C, 100 rpm on orbital shaker), in 200 mL of basic mineral medium (g/L): KH_2_PO_4_ 3.4; K_2_HPO_4_ 4.4; NaNO_3_ 15; KCl 1.1; NaCl 1.1; MgSO_4_ 0.224; yeast extract 0.5; (mg/L): ZnSO_4_·7H_2_O 1.45; CuSO_4_·5H_2_O 1.25; MnSO_4_·H_2_O 8.4; CaCl_2_·4H_2_O 1.2; FeSO_4_ 0.28. Cultivations were carried out until stationary phase and maximum rhamnolipid concentration (highest rhamnolipid concentration observed in given experiment) was reached. The time in which the maximum rhamnolipid concentration was observed was defined as maximum production time. Biomass growth and rhamnolipid determination was regularly detected with sampling.

For the experiments exploring the effect of the type and amount of carbon source on rhamnolipid production, following carbon sources were added to the medium (separately after autoclaving): sodium citrate, glycerol, succinic acid, FAME (fatty acids methyl esters), sunflower oil, or hexadecane in concentrations 5, 10, 20, 40 g/L. In the experiments studying the type of carbon source, the concentration of nitrogen source was constant; 15 g/L NaNO_3_.

The biomass growth was determined as dry weight at 105 °C after centrifugation. The biomass growth was not determined for the hydrophobic carbon sources (FAME, sunflower oil, hexadecane) due to the formation of emulsion caused by the surfactant production, which did not allow the separation of cells from the carbon source. All experiments were performed in triplicate independent experiments, error bars in figures represent standard deviation.

### 2.3. Response Surface Methodology (RSM) Optimization of Culture Medium

The surface response methodology was used for the design of the experiment determining the effect of glycerol and NaNO_3_ concentration on the rhamnolipid production. The concentration ranges of these two variables (see Table 1) were estimated according to screening tests and literature. The central composite design (CCD) utilizing the Design expert 9.0 software (Stat-Easy Inc., Minneapolis, MN, USA) was used to generate 13 experiments (see Section 3.2). These experiments were performed as described in Section 2.2, rhamnolipid concentration and maximum production time were determined.

From the experimental data, a second-order polynomial regression model equation was derived:Y = α1 + α2X_1_ + α3X_2_ + α4X_1_X_2_ + α5X_1_^2^+ α6X_2_^2^
where Y is the predicted response value (both rhamnolipid yield and rhamnolipid productivity were assessed); X_1_ (glycerol concentration) and X_2_ (NaNO_3_ concentration) represent variable factors; α_n_ coefficients were determined from the regression results.

### 2.4. Rhamnolipid Determination and Isolation

The total rhamnolipid content was determined as the concentration of rhamnose by the phenol–sulphuric method [18] with rhamnose as standard. The validity of phenol–sulphuric method accuracy for the studied rhamnolipid determination was verified via HPLC after acidic rhamnolipid hydrolysis as rhamnose according to [19]. After determining the rhamnolipids composition by MS, correlation factor between rhamnose and total rhamnolipids was estimated and used for the experiment evaluation. The rhamnolipid yield (g/L) was determined as the maximum cumulative rhamnolipid production during the cultivation. The overall productivity of rhamnolipid (mg_rhamnose_/L/h) was determined as the cumulative rhamnolipid production (mg/L)/culture time (h). Rhamnolipids were isolated as described previously [20]. Briefly, the biomass was removed by centrifugation (10,000× *g*, 15 min) and the supernatant was subjected to acidic precipitation (1M HCl, 4 °C, 24 h). The precipitate was centrifuged (10,000× *g*, 30 min) and extracted five times by a chloroform:methanol solution (1:1) and analyzed by MS.

### 2.5. Mass Spectrometry

The rhamnolipids were analyzed via LTQ OrbitrapVelos mass spectrometer (Thermo Scientific, Waltham, MA, USA) according to the previously published procedure [21]. Briefly, a high resolution FT negative ion mode was employed for the resolution hybrid tandem MS (MS^2^). MS spectra were acquired with R = 30,000 at m/z 400 (scan range m/z 150–2000). ESI-MS mode was used with ion spray voltage—2500 V. The nebulizer gas employed was nitrogen with parameters of the sheath gas 18 arbitrary units and auxiliary gas set at seven arbitrary units. CID experiments were performed with helium as a collision gas at normalization energy of 35% for the fragmentation of the parent ions.

### 2.6. Critical Micelle Concentration

The critical micelle concentration (CMC) was determined by contact angle measurement [22,23]. This method is based on an inverse relation of surface tension and the contact angle of the liquid (surfactant solution) placed on a solid surface. For CMC determination, the isolated rhamnolipids were diluted in deionized water and a series of solutions with rhamnolipid concentration ranging from 1 to 100 mg/L were prepared. A 5 µL drop of a rhamnolipid solution was placed on a polystyrene plate (constant temperature 25 °C) and the high-resolution camera was used to acquire images. Each measurement was repeated at least 10 times. The contact angle was determined by the CAM 2008 software (KSV Instrument, Espoo, Finland). CMC was obtained from a plot of contact angle as a function of rhamnolipids concentration. The break point observed in the plot corresponded to the value of CMC.

### 2.7. Determination of Emulsification Activity

For the surfactant emulsification activity, a previously published procedure with modification as follows was used [24]. Briefly, 1.2 mL of a hydrocarbon or an oil (hexane, crude petroleum, sunflower oil) and 0.8 mL of a surfactant (1 g/L) were vortexed at high speed for 2 min in a tube. The mixture was left to stabilize for 24 h without shaking. The emulsification index was determined by measuring the height of the emulsion layer, dividing it by the total height of the mixture and multiplying the resulting value by 100. Each measurement was performed in five independent replicates, error bars represent standard deviation.

### 2.8. Oil Displacement Test

The oil displacement test was performed by adding 100 mL of distilled water to a Petri dish (diameter 25 cm). After that, 70 µL of crude petroleum was dropped onto the water surface, followed by addition of 20 µL of aqueous solution of a surfactant (concentration 1 g/L) onto the surface of the oil. The diameter of the clear zone indicates the surface tension reduction efficiency of a given surfactant [24]. Each measurement was performed in five independent replicates.

### 2.9. Solubilization of Phenanthrene

The solubility of phenanthrene in the presence of a biosurfactant was determined using a batch test [25]. Phenanthrene (2 mg) was added to 10 mL of deionized water (in excess amount compared to its solubility) and various concentrations of the biosurfactants were added. The biosurfactant concentrations spanned over a wide range of values below and above CMC (0–150 mg/L). The samples were placed on an orbital shaker at 20 °C and 100 rpm. After 48 h of shaking, the samples were left for 24 h without shaking and the supernatants were extracted by hexane (supernatant:hexane 2:1). The hexane extract was evaporated and the extracted samples were dissolved in 1 mL of acetonitrile and analyzed for phenanthrene by high performance liquid chromatography (Agilent 1100 Series, Agilent Technologies Inc., Santa Clara, CA, USA), column: Watrex 125X4 Nucleosil C8, acetonitrile/water 70/30) with a DAD detector.

## 3. Results and Discussion

### 3.1. Effect of Carbon and Nitrogen Source on Rhamnolipid Production

Cultivation conditions can significantly affect rhamnolipid production. Type, concentration and ratio of carbon and nitrogen sources (C/N ratio) are among the most frequently studied factors [26]. We designed the initial tests verifying the effect of these factors on the production of rhamnolipids by *Pseudomonas aeruginosa* DBM 3774 according to previously published procedure [21]. Six different carbon sources were used (hydrophilic—sodium citrate, glycerol, succinic acid; hydrophobic—sunflower oil, FAME hexadecane) and NaNO_3_ (15 g/L) was employed as nitrogen source. The nitrogen concentration was based on previous experiments (see [21]) as suitable for cultivation in the experimental set-up. It is important to note that cultivation conditions in Erlenmeyer flasks and bioreactor can vary significantly with respect to culture conditions (oxygen transfer, homogenization) and thus the results of screening experiments (for which are Erlenmeyer flasks suitable) are a preliminary step for scale-up procedures. The results indicate that glycerol was the most effective carbon source for rhamnolipid production (Table 2). The highest concentration of glycerol (40 g/L) yielded the highest amount of rhamnolipids (4.37 g_rhamnose_/L), with productivity reaching 22.66 mg_rhamnose_/L/h. This fact corresponds with the results reported previously [27], where the highest production of rhamnolipids (3.0 g/L) was observed in the presence of combination of glycerol and NaNO_3_. Vegetable oils are frequently used as a carbon source for rhamnolipid production by *Pseudomonas aeruginosa* strains [6,28]. However, in our experiment, sunflower oil as a C-source (40 g/L) occupied the second place with a production reaching 2.61 g_rhamnose_/L and productivity 13.57 mg_rhamnose_/L/h. Hexadecane was found to be an unsuitable carbon source, although alkanes are often mentioned as enhancers of rhamnolipid production [29]. Based on the initial tests, glycerol was selected for further studies of the carbon source concentration and C/N ratio effect on the rhamnolipid production using the RSM.

### 3.2. Optimization of Culture Medium for Rhamnolipid Production by RSM

The optimal amount and ratio of carbon and nitrogen source is a prerequisite for efficient rhamnolipid production [26]. Hence, we aimed to study the conditions that would improve the rhamnolipid production by *Pseudomonas aeruginosa* DBM 3774 under studied conditions by evaluation the effect of various concentrations and ratio of carbon and nitrogen source with the use of the RSM. The verification of the usefulness of the RSM model and its results will allow to build a robust methodic approach for rhamnolipid production scale-up. The macronutrient concentrations were tested in the range of 0–50 g/L for glycerol and 0–15 g/L for NaNO_3_. These ranges pose a C/N ratio from 1/1.75 to 19/1. As results reported in literature state nitrogen limiting conditions as beneficial for rhamnolipid production [13], we have also explored additional experiments outside the suggested RSM values with C/N ratio 52/1 and 130/1 (glycerol 40 g/L as carbon source) with results confirming that the *P. aeruginosa* DBM 3774 under studied conditions does not improve rhamnolipid production at severe nitrogen limitation, as the obtained rhamnolipid yield was 2.6 and 2.3 g_rhamnose_/L, respectively, therefore significantly lower than the value 4.37 g_rhamnose_/L found at NaNO_3_ 15 g/L.

The RSM was used to which parameters would be the most suitable for the model to establish and help to investigate the effect of these components on rhamnolipid production by *Pseudomonas aeruginosa* DBM 3774 under batch cultivation conditions. Standard performance indexes, the rhamnolipid yield and rhamnolipid productivity, were chosen as the response values for the RSM. The applicability of RSM model in biotechnology needs to be verified for each model application, as the response of the microbial producer can vary not only between microbial types but the specific cultivation set-up as well. To establish a minimal design for two independent variables resulting in suitable number of test runs, design utilizing five coded variables levels (Table 1) were used, resulting in 13 test runs. The experimental design runs and obtained experiment results are showed in Table 3. This RSM model was confirmed to correspond sufficiently to resulting rhamnolipid production while maintaining appropriate number of test runs which would be applicable even in larger scale cultivations.

According to the response values obtained from the design experiment, a second- order equations were generated, as follows:Rhamnolipid yield (g/L) = −2.59719 + 0.89848X_1_ + 1.43528X_2_ + 9.49333·10^−3^X_1_X_2_ − 0.011656X_1_^2^ − 0.089249X_2_^2^
Rhamnolipid productivity (mg/L/h) = 0.16962 + 0.084441X_1_ + 0.24785X_2_ + 6.80000·10^−3^X_1_X_2_ − 9.92400·10^-4^X_1_^2^ − 0.016182X_2_^2^

The X_1_ and X_2_ indicate the two variables (glycerol concentration and NaNO_3_ concentration, respectively).

A total of 13 batch experiments was generated by the RSM and performed according to 2.2. The regression models (see Section 3.2) fitted experimental data very well with high R^2^ values of 0.9994 and 0.9983, respectively. The three dimensional response surfaces based on the regression models are plotted in Figure 1A,B. The highest point of the 3D plot indicates the conditions (concentrations of carbon and nitrogen source) that should result in the highest predicted rhamnolipid yield (Figure 1A) and productivity (Figure 1B). According to the prediction model of Figure 1A, the rhamnolipid yield should increase with increasing concentrations of both macronutrients and therefore the best performance for the rhamnolipid yield would be outside the selected range of variables. However, the rhamnolipid yield variable does not reflect the time parameter (which is an important factor for process economics), nor does this model include the phenomenon of substrate inhibition, therefore the model of rhamnolipid productivity was followed in consequent experiments. The regression model of rhamnolipid productivity (Figure 1B) predicted the highest productivity value (24.0 mg_rhamnose_/L/h) at concentrations of 42.7 g/L for glycerol and 10.3 g/L for NaNO_3_ (C/N ratio 11.48). According to the models, the concentration of glycerol had more significant effect on the productivity than the concentration of NaNO_3_.

### 3.3. Experimental Verification of the RSM Model

The concentrations predicted as ideal by the RSM for rhamnolipid productivity were tested in a flask cultivation (glycerol 42.7 g/L, and NaNO_3_ 10.3 g/L). The predicted optimal concentrations were found out similar to the arbitrarily assigned values in the initial experiments performed to evaluate the carbon source (i.e., glycerol 40 g/L, and NaNO_3_ 15 g/L, see Table 2) and therefore the maximum productivity was also close in value. The experiment verification of the RSM predicted optimal values showed final rhamnolipid yield (5.6 g_rhamnose_/L) and rhamnolipid productivity (22.85 mg_rhamnose_/L/h), which reached 97.56% and 95.2% of the predicted values, respectively. The difference between predicted and experimental data is not significant in terms of the error determination. Using the correlation coefficients obtained in Section 3.4.1 we can conclude that the final rhamnolipid yield reached 13.5 g/L (total rhamnolipids) and productivity 45.3 mg/L/h (total rhamnolipids). These results indicate that the RSM can be effectively used in the design of an optimal medium for rhamnolipid production. Under similar conditions rhamnolipid productivity was reported previously at 23.2 mg_rhamnolipid_/L/h [27]. The good correspondence of predicted values and obtained results for rhamnolipid productivity based on the utilized RSM model with coded variable levels and test runs (Table 2) verified the possibility of this experimental set-up for prediction of cultivation parameters of *P. aeruginosa* DBM 3774.

### 3.4. Rhamnolipid Characterization

Carbon and nitrogen sources have important effect on rhamnolipid production and the characteristics of the produced rhamnolipid mixture including the content of rhamnolipid congeners [12]. To elucidate the effect of the carbon and nitrogen source studied in this work, we have determined the composition of congeners and basic physico-chemical characterization of rhamnolipid mixture obtained from RSM predicted medium with (42.7 g/L glycerol, 10.3 g/L NaNO_3_) and a control medium using hydrophilic substrate with the highest rhamnolipid yield other than glycerol (20 g/L sodium citrate, 15 g/L NaNO_3_, see Table 2).

#### 3.4.1. Structural Characterization of Rhamnolipids

The rhamnolipids obtained from the cultivations on the RSM predicted and on the control media, were examined by tandem mass spectrometry to ascertain the rhamnolipid congener composition (Table 4). Significant differences between the two studied rhamnolipid mixtures were found. The results showed that the changes in the carbon source and C/N ratio, beside improving the production of rhamnolipids, also significantly modified the qualitative composition of the resulting rhamnolipids. Four classes of rhamnolipid congeners (monorhamno-monolipids RhaFA, monorhamno-dilipids RhaFAFA, dirhamno-monolipids RhaRhaFA and dirhamno-dilipids RhaRhaFAFA) were found in both studied mixtures (see Table 4). Although in some strains, HAAs were detected as part of the rhamnolipid mixture [5,30] in rhamnolipids produced by *P. aeruginosa* DBM 3774 and other non-pseudomonas microorganisms, these were not found by the tandem mass spectrometry method employed in detectable amounts [31]. The rhamnolipids obtained from the cultivation under RSM predicted conditions belonged primarily to the RhaFAFA class (49.9%), while the rest was distributed almost evenly between the other three classes. Rhamnolipids from the control medium were mostly representatives of two classes–RhaFAFA and RhaRhaFAFA (50.2% and 47.3%, respectively). However, in both groups, the decanoic acid was the major FA. The tandem mass spectrum of most abundant rhamnolipid (RhaRha1010) is shown as Appendix A. These findings are in accordance with published data, which state that rhamnolipids produced by *Pseudomonas aeruginosa* usually display a low variability in fatty acid composition and that the decanoic fatty acid is the most common among them [9,20]. A more detailed analysis of the rhamnolipid congeners in both mixtures implied a possibility of the cultivation medium components having an influence on the complex biosynthetic pathways for rhamnolipid synthesis in *P. aeruginosa*. The proteome of *Pseudomonas aeruginosa* PA1 has been studied with respect to the rhamnolipid biosynthesis [32]. It was demonstrated that culture media differing in the sources of carbon and nitrogen (stimulating or inhibiting rhamnolipid synthesis) have led to significant changes in the proteome of both cell populations. These changes included proteins closely related to the biosynthesis of rhamnolipids. In our experiments, the RSM predicted medium containing glycerol as the sole carbon source and having a high C/N ratio yielded rhamnolipids with a significantly higher proportion of “simpler” rhamnolipid congeners in comparison with the product formed on the control medium with citrate as the sole carbon source and a balanced C/N ratio (Table 4). The rhamnolipid mixture obtained by cultivation on RSM predicted media had a significantly greater proportion of rhamnolipid congeners containing one rhamnose or one fatty acid or a combination of them—one rhamnose and one fatty acid.

A short fatty acid (octanoic acid) was also found in the RSM predicted medium produced rhamnolipids in significant quantities. In contrast, the control medium gave higher proportion of congeners containing two rhamnoses, two fatty acids or a combination thereof. Moreover, only the rhamnolipid mixture from control medium contained congeners with unsaturated fatty acids, specifically decenoic and dodecenoic acid. These considerable differences in the rhamnolipid mixtures composition did not significantly affect their surface activity and the other physical properties (see below). Because of the specific importance of rhamnolipids for the producer itself [20], the question is, whether the producer is able to respond to the external environment (media composition) by changing the rhamnolipid biosynthesis strategy so that the resulting product had the most suitable properties.

Based on the structural characterization, correlation coefficients between rhamnose and rhamnolipid content were determined. The correlation coefficients for rhamnolipid mixtures were 2.41 and 2.52 for RSM predicted medium and control medium, respectively.

#### 3.4.2. Critical Micelle Concentration

Critical micelle concentration is one of the most important physico-chemical characteristic of surfactants. CMCs of the rhamnolipid mixtures produced by *Pseudomonas aeruginosa* DBM 3774 were investigated by the contact angle measurement. Although the optimization of the culture medium had significant effect on the rhamnolipid mixture composition, the CMC values were similar for both rhamnolipid mixtures, namely 90.5 mg/L for the RSM predicted medium and 85.1 mg/L for the control medium.

In studies concerning the relationship between the rhamnolipid congener composition and the CMC, several effects of the fatty acid chain length and saturation were described. The occurrence of longer fatty acids in rhamnolipids leads to the formation of micelles at a lower concentration (demonstrated as lower CMC) [33]. In this study, a rhamnolipid mixture rich in RhaRhaC10C12:1 and RhaRhaC10C12 had a lower CMC than a mixture with high content of RhaRhaC10C10. Our results are in agreement with their results, although the relative difference in the CMCs was small in our case. The rhamnolipid mixture from the control medium provided a lower CMC because it had a high content of rhamnolipids with longer fatty acids (C12; 36%). The mixture from the RSM predicted medium had a higher CMC due to greater amount of short fatty acids (C8; 35%).

In addition, the mixture from the control medium (with lower CMC) contained a greater amount of the congener RhaRhaC10C10 (26.24%), in comparison to the mixture from the RSM predicted medium, where only 8.7% of this homologue was found. The lower CMC of rhamno-dilipids (in comparison with rhamno-monolipids) has been reported [29]. In this study, this effect was demonstrated by determining the CMC of isolated congeners and RhaRhaC10C10 provided significantly lower CMC (5 mg/L) in comparison to RhaRhaC10 (CMC 200 mg/L).

The CMC is also significantly affected by the fatty acid saturation, as was proved in previous studies [34,35]. In these studies, the CMC values increased with the increasing content of rhamnolipid congeners containing unsaturated fatty acids. This fact could offset the effect of longer fatty acids and the high content of RhaRhaC10C10 in the mixture from the control medium in our experiments. The rhamnolipids from the control medium contained 18.8% of rhamnolipid congeners with at least one unsaturated fatty acid, contrarily to the mixture from the RSM predicted medium, which consisted only of saturated fatty acids. A combination of all the stated effects gives a probable explanation of the similarity of CMC values of the two studied rhamnolipid mixtures.

#### 3.4.3. Emulsification Index

Emulsification activity against hydrophobic compounds is an important characteristic of biosurfactants, used for evaluation of their potential applicability in the industry (medicine, cosmetic industry, remediation technologies) [36]. Model substrates were used for the determination of the emulsification index: alkane (hexane), vegetable oil (sunflower oil), and crude petroleum. Both rhamnolipid mixtures exhibited a good emulsification activity against all the tested substrates (Figure 2). The best result was achieved in the case of sunflower oil, where the emulsion stabilization by both rhamnolipid mixtures reached up to 70%.

The rhamnolipid mixture from the RSM predicted medium had a slightly better emulsification activity against hexane (55. 9%) and crude oil (54.5%) than the mixture from the control medium (emulsification indexes 47.3% and 50.8%, respectively). For rhamnolipids produced by *Pseudomonas aeruginosa* TMN on glycerol were reported similar results (emulsification index 50% against petroleum product—kerosene) [37].

#### 3.4.4. Oil Displacement Test

The oil displacement test is an indirect indicator of the activity of a surfactant against a specific oil. A larger diameter of the clear zone means a higher surface activity of the surfactant [38]. In this study, crude petroleum was chosen as a model hydrophobic compound. Both rhamnolipid mixtures provided similar results. Namely, the rhamnolipids from the RSM predicted medium generated a clear zone with diameter 11.5 cm and the rhamnolipids from the control medium formed a clear zone with diameter 11.2 cm. The oil displacement test was performed with Tween 80 as a synthetic surfactant control. The obtained clear zone 3.8 cm for Tween 80 proved a significantly better surface tension reduction efficiency of both rhamnolipid mixtures, confirming the observation that biosurfactants have better physico-chemical properties than synthetic surfactants in the same conditions [39].

#### 3.4.5. Phenanthrene Solubilization

PAHs are one of the most frequently occurring encountered persistent organic pollutants in oil-contaminated wastewaters. Their bioavailability for microorganisms is limited by their low water solubility. Addition of biosurfactants can enhance the relative solubility of the PAHs in water and thus their bioavailability [40], although the effect is dependent on the specific microbial compatibility with applied biosurfactant. Liu*,* et al. [41] focused on the relation between solubility and bioavailability of hexadecane for *P. aeruginosa* (rhamnolipid producer) and *P. putida* (non-producer). The bioavailability of solubilized hexadecane for *P. aeruginosa* was significantly enhanced but in the case of *P. putida*, the opposite phenomenon was observed. It is likely that rhamnolipid layer at the organic-water inter-face has a blocking effect on the bioavailability for *P. putida* due to its innate inability to uptake hexadecane-rhamnolipid aggregates as opposed to natural rhamnolipid producer such as *P. aeruginosa*.

Figure 3 shows the enhanced solubilization of phenanthrene by both rhamnolipid mixtures. The solubilization effect was dependent on the concentration of surfactant and the rhamnolipid mixture origin. The rhamnolipid mixture obtained from the RSM predicted medium provided a higher solubilization effect at all tested concentrations. The highest solubilization effect of phenanthrene (32.1 mg/L) by the mixture from the RSM predicted medium was observed at a concentration of 200 mg/L (the highest tested concentration). When the rhamnolipid concentration exceeded 50 mg/L, a rapid increase in the solubility of phenanthrene was observed. Similar turning point at a rhamnolipid concentration of 50 mg/L was reported previously [42].

## 4. Conclusions

The possibilities of enhancing rhamnolipid production by *Pseudomonas aeruginosa* DBM 3774 using media composition design and resulting rhamnolipid properties were studied. The applicability of RSM prediction was verified for media composition optimization with significant increase in the rhamnolipid yield and productivity in batch cultivation. The rhamnolipid productivity was found as more suitable parameter for RSM model based prediction than rhamnolipid yield.

The most interesting results were found in comparing the rhamnolipid mixture composition and physico-chemical properties of rhamnolipid mixtures obtained under different cultivation conditions, which were found to differ in some significant aspects. Although mass spectrometry revealed that substitution of sodium citrate by higher concentration of glycerol in the culture medium led to the production of rhamnolipid mixture comprising only of saturated FAs, physico-chemical properties of both mixtures were similar.

## Figures and Tables

**Figure 1 microorganisms-10-01272-f001:**
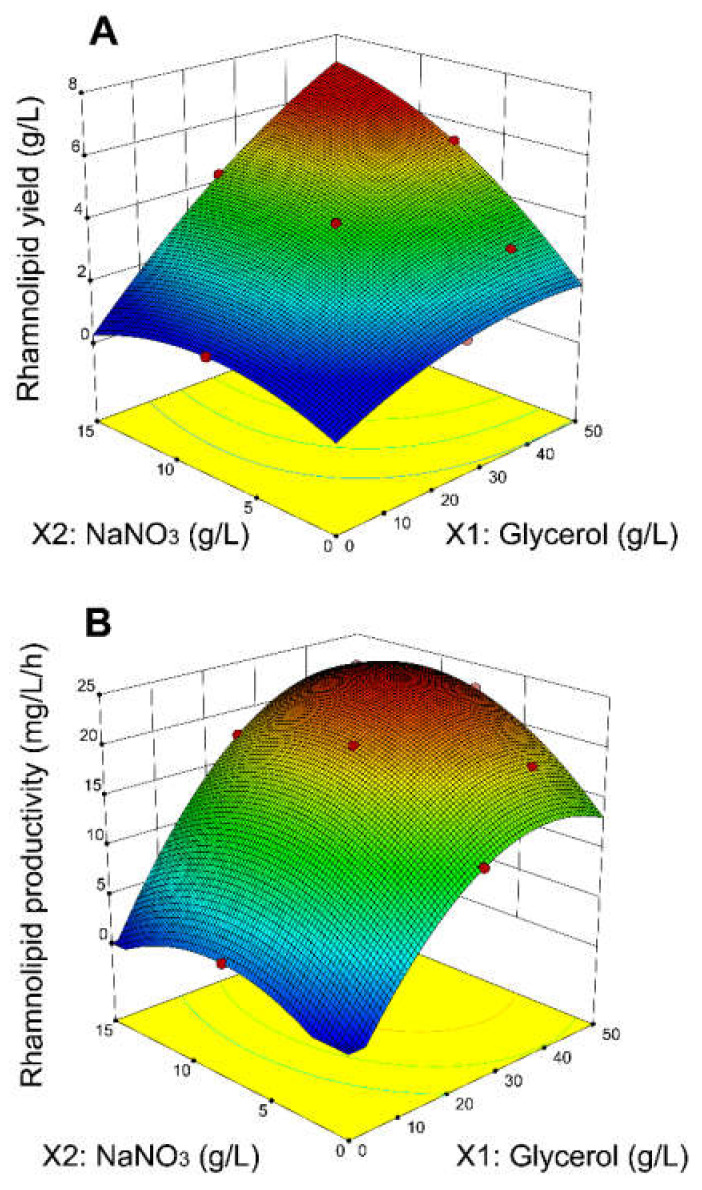
3D surface plots obtained from the regression models based on the RSM designed experiments, showing the effect of glycerol (X1) and NaNO_3_ (X2) on the rhamnolipid yield (**A**) and rhamnolipid productivity (**B**).

**Figure 2 microorganisms-10-01272-f002:**
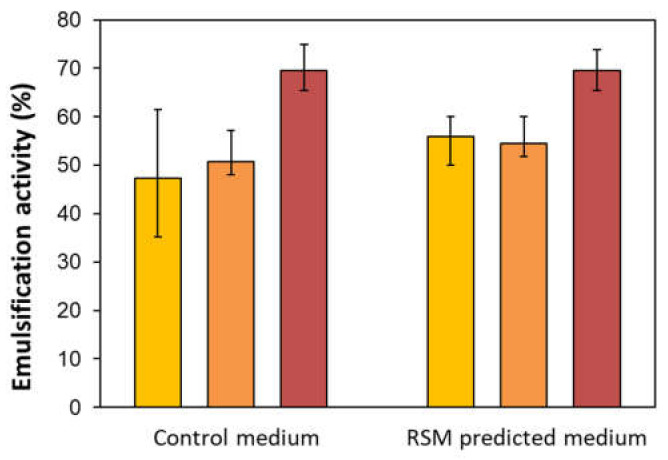
Emulsification activity of the rhamnolipid mixtures obtained from cultivation of *Pseudomonas aeruginosa* DBM 3774 in the RSM predicted and the control medium: (■) hexane, (■) crude oil, (■) sunflower oil. Error bars represent standard deviation.

**Figure 3 microorganisms-10-01272-f003:**
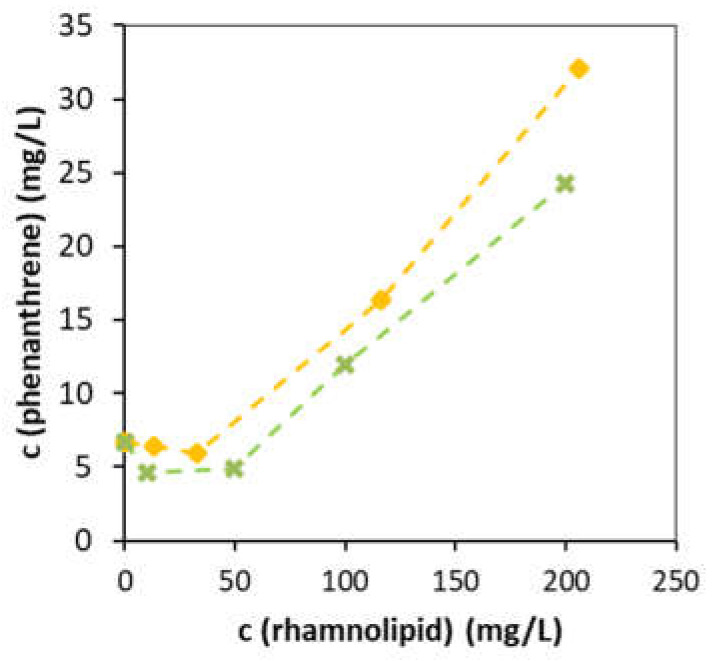
Phenanthrene solubilization with rhamnolipid mixtures produced by *Pseudomonas aeruginosa* DBM 3774 on different medium composition. (■) rhamnolipid from the RSM predicted medium, (**×**) rhamnolipid from the control medium.

**Table 1 microorganisms-10-01272-t001:** Coded levels of variables (glycerol, NaNO_3_), their corresponding specific levels and concentration range, used for the response surface methodology design.

Coded Levels of Variables					
Variable	−1.41	−1	0	1	1.41
Glycerol (g/L)	0	7.3	25	42.7	50
NaNO_3_ (g/L)	0	2.2	7.5	12.8	15

**Table 2 microorganisms-10-01272-t002:** Effect of carbon source on the rhamnolipid production by *Pseudomonas aeruginosa* DBM 3774 (expressed as rhamnose content) (nitrogen source: NaNO_3_ 15 g/L).

	Carbon Source Concentration	Biomass	Max. Production Time	Yield	Productivity
Carbon Source	(g/L)	(g/L)	(h)	(g_rhamnose_/L)	(mg_rhamnose_/L/h)
sodium citrate	5	0.39	193	0.43	2.20
	10	0.73	241	0.47	1.95
	20	1.43	193	0.93	4.82
	40	0.87	143	0.70	4.91
glycerol	5	0.86	193	0.83	4.30
	10	1.61	168	1.64	9.76
	20	1.68	221	3.16	14.30
	40	1.88	193	4.37	22.66
succinic acid	5	0.40	221	0.33	1.48
	10	0.97	241	0.63	2.61
	20	1.00	193	0.45	2.33
	40	0.60	193	0.43	2.23
FAME	5	nd	145	0.44	3.03
	10	nd	168	0.89	5.32
	20	nd	234	2.23	9.53
	40	nd	192	2.31	12.01
sunflower oil	5	nd	192	1.17	6.08
	10	nd	192	1.51	7.86
	20	nd	234	2.16	9.24
	40	nd	192	2.61	13.57
hexadecane	5	nd	168	0.24	1.42
	10	nd	168	0.26	1.53
	20	nd	168	0.26	1.55
	40	nd	145	0.31	2.11

nd—not determined.

**Table 3 microorganisms-10-01272-t003:** RSM designed matrix of independent variables and their corresponding experimental values for rhamnolipid yield and productivity by *Pseudomonas aeruginosa* DBM 3774.

	Glycerol		NaNO_3_		Max. Production Time	Rhamnolipid Yield	Rhamnolipid Productivity
Run	Coded Variable	c (g/L)	Coded Variable	c (g/L)	(h)	(g_rhamnose_/L)	(mg_rhamnose_/L/h)
1	0	25	0	7.5	194	3.90	20.10
2	1	42.7	−1	2.2	169	3.07	18.13
3	−1	7.3	1	12.8	237	1.89	7.99
4	−1.41	0	0	7.5	357	1.12	3.14
5	1	42.7	1	12.8	264	6.20	23.45
6	0	25	−1.41	0	132	1.66	12.58
7	0	25	1.41	15	244	4.29	17.59
8	1.41	50	0	7.5	241	5.41	22.49
9	−1	7.3	−1	0	210	1.31	6.23
10	0	25	0	7.5	195	3.92	20.15
11	0	25	0	7.5	193	3.87	20.08
12	0	25	0	7.5	194	3.91	20.11
13	0	25	0	7.5	190	3.82	20.08

**Table 4 microorganisms-10-01272-t004:** MS^2^ analysis of RhaRhaFAsFAs, RhaFAsFAs, RhaFAs and RhaRhaFAs fractions of *Pseudomonas aeruginosa* DBM 3774 rhamnolipids cultivated in the RSM predicted and the control medium.

	RSM Medium	Control Medium		RSM Medium	Control Medium
*RhaRhaFAFA*			*RhaFAFA*		
RhaRhaC8C8	3.0	0.0	RhaC8C8	8.0	0.02
RhaRhaC8C10	2.2	2.0	RhaC8C10	5.4	2.9
RhaRhaC10C8	4.6	1.5	RhaC10C8	4.5	2.0
RhaRhaC10C10	8.7	26.2	RhaC10C10	23.5	25.9
RhaRhaC10C10:1	0.0	0.2	RhaC10C10:1	0.0	0.3
RhaRhaC10:1C10	0.0	0.2	RhaC10:1C10	0.0	0.2
RhaRhaC10C12:1	0.0	4.1	RhaC10C12:1	0.0	4.0
RhaRhaC12:1C10	0.0	5.7	RhaC12:1C10	0.0	3.4
RhaRhaC10:1C12:1	0.0	0.0	RhaC10:1C12:1	0.0	0.01
RhaRhaC10C12	1.0	5.6	RhaC10C12	4.0	3.7
RhaRhaC12C10	0.4	4.0	RhaC12C10	3.5	4.5
RhaRhaC12C12	0.0	0.3	RhaC12C12	1.0	0.1
RhaRhaC12:1C12:1	0.0	0.02	RhaC12:1C12:1	0.0	0.02
RhaRhaC12C12:1	0.0	0.4	RhaC12C12:1	0.0	0.2
total *RhaRhaFAFA*	19.9	50.2	total *RhaFAFA*	49.9	47.3
*RhaFA*			*RhaRhaFA*		
RhaC8	3.2	0.03	RhaRhaC8	4.1	0.1
RhaC10:1	0.0	0.03	RhaRhaC10:1	0.0	0.1
RhaC10	9.6	0.4	RhaRhaC10	9.2	1.7
RhaC12	2.3	0.02	RhaRhaC12:1	0.0	0
RhaC12:1	0.0	0	RhaRhaC12	1.8	0.06
total *RhaFA*	15.1	0.5	total *RhaRhaFA*	15.1	2.0

## Data Availability

Not applicable.

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
