# Peer review of "Production and Characterization of Rhamnolipids Produced by Pseudomonas aeruginosa DBM 3774: Response Surface Methodology Approach"

_microorganisms, 2022, doi:10.3390/microorganisms10071272_

Round 1
Reviewer 1 Report
Attached file

Reviewer 2 Report
This study is about optimization of microbial lipid production generated by P. aeruginosa. There are not so new things, so they need to add some new insights or new results. Major revision.
1. Figures are not interesting. Please add pictures of experimental setting and reactor.
2. Improve your introduction by showing other related technology. For example, P. aeruginosa can be also used in microbial electrochemical systems for bioelectricity generation (J Korean Soc Environ Eng 2020 Jul; 42(07): 360-380, Chem Eng J 2021 Nov; 424: 130388)
3. Figure quality is not good. Please improve them. Especially, letters in them are so small so that they are not so well readable.
4. Data points are so small for optimization.
5. Overall, manuscript writing should be improved. Emphasize core findings.
Round 2
Reviewer 1 Report
The main points raised by this reviewer were properly addressed.
Just two additional minor corrections are suggested:
Line 41 "....the main focus in on the production..." to "...the main focus is on the production...".
Line 231 "...at sever nitrogen limitation..." to "...at severe nitrogen limitation...".
Author Response
Reviewer: Line 41 "....the main focus in on the production..." to "...the main focus is on the production...".
Reply: The mistake has been corrected.
Line 231 "...at sever nitrogen limitation..." to "...at severe nitrogen limitation...".
Reply: The mistake has been corrected.
Reviewer 2 Report
It is acceptable.
Author Response
Comments and Suggestions for Authors: It is acceptable.Reply: We thank the reviewer for the comment.